# Methyl groups as widespread Lewis bases in noncovalent interactions

Oliver Loveday [1] & Jorge Echeverría [1,2 ✉]

It is well known that, under certain conditions, C($sp^3$) atoms behave, via their σ-hole, as Lewis acids in tetrel bonding. Here, we show that methyl groups, when bound to atoms less electronegative than carbon, can counterintuitively participate in noncovalent interactions as electron density donors. Thousands of experimental structures are found in which methyl groups behave as Lewis bases to establish alkaline, alkaline earth, triel, tetrel, pnictogen, chalcogen and halogen bonds. Theoretical calculations confirm the high directionality and significant strength of the interactions that arise from a common pattern based on the electron density holes model. Moreover, despite the absence of lone pairs, methyl groups are able to transfer charge from σ bonding orbitals into empty orbitals of the electrophile to reinforce the attractive interaction.

[1] Departament de Química Inorgànica i Orgànica, Universitat de Barcelona, Barcelona, Spain. [2] Institut de Química Teòrica i Computacional IQTC-UB, Universitat de Barcelona, Barcelona, Spain. ✉email: jorge.echeverria@qi.ub.es

Generally, noncovalent interactions appear when an electron-rich and an electron-depleted species, i.e., the Lewis base and acid, respectively, come close to each other[1]. The energy stabilization of the adduct with respect to the isolated molecules has been traditionally explained in terms of electrostatics and/or orbital interactions[2]. Electrostatic attraction was first rationalized by Politzer et al. under the electron density holes model, which has been significantly studied and expanded ever since[3–5]. Orbital interactions, on the other hand, are based on charge transfer processes from an occupied orbital (e.g., but not necessarily, a lone pair) into an empty antibonding orbital. Usually, noncovalent interactions are the combination of the two effects along with dispersion forces[6–8].

It has been accepted in recent years that the name of a noncovalent interaction is given by the nature of the Lewis acids. For instance, atoms from groups 14, 15, 16, and 17 acting as Lewis acids give rise to tetrel, pnictogen, chalcogen, and halogen bonding, respectively, although the physical origin of the attraction is practically the same in all cases: the presence of a region of electron depletion, the σ-hole. At the other end of the interacting unit, the most common Lewis bases are anions and lone pair-containing molecules. However, other electron-rich species have been seen to be capable to act as electron density donors as, for example, C=C double bonds and π-conjugated systems. In general, sp and sp²-hybridized carbon atoms are considered good Lewis bases and one can find in the literature many examples of them involved in noncovalent interactions[9]. In some cases, carbon atoms in carbenes can also act as Lewis bases[10–14].

On the other hand, $sp^3$ carbon atoms are generally found acting as Lewis acids in tetrel bonding[15–21]. More rarely, $sp^3$ carbon atoms have been described as electron density donors[22]. The term ditetrel bond has been recently coined by Scheiner to name the interaction between positively and negatively charged tetravalent group 14 atoms[23], an interaction pattern that was previously seen in substituted silane-methane adducts[24]. Methyl groups acting as electron density donors in π-hole bonding have also been reported in the crystal structures of trimethylgallium and trimethylindium[22]. Furthermore, the methane–water system, which can display a configuration involving a short $H_4C\cdots H$-OH contact, has been extensively investigated by means of laser spectroscopy[25], microwave spectrometry[26], and also theoretically[27–30]. The $CH_4\cdots H_2O$ adduct has been proposed[31] as a precursor of $CH_5^+$, with a calculated binding energy of 1 kcal mol$^{-1}$, increasing up to 10 kcal mol$^{-1}$ in the case of $CH_4\cdots H_3O^+$[32]. Similar interactions have been computationally predicted for the adsorption of water on $CH_3$:Si(111)[33].

In light of this, a question arises: is this behavior reduced to computationally studied hypothetical systems, the methane–water adduct and one particular family of experimental structures? In this work, we show that methyl groups, despite not having an available lone pair, are extensively found behaving as Lewis bases in practically all known major types of noncovalent interactions, including triel, tetrel, pnictogen, chalcogen, and halogen bonding.

## Results and discussion

**Molecular electrostatic potential**. To understand the origin of such behavior, the capability of methyl groups to act as electron density donors needs to be evaluated and the requirements for a $sp^3$ carbon atom to be electron-rich must be understood. A straightforward way to calibrate this capability is the plotting of the molecular electrostatic potential (MEP) around the methyl group. It is known that common electron-withdrawing substituents such as oxygen, sulfur, or halogen atoms create an electron density depletion region at the carbon atom, the σ-hole, which is clearly visible in MEP maps. On the other hand, one could expect that electropositive atoms (or at least less

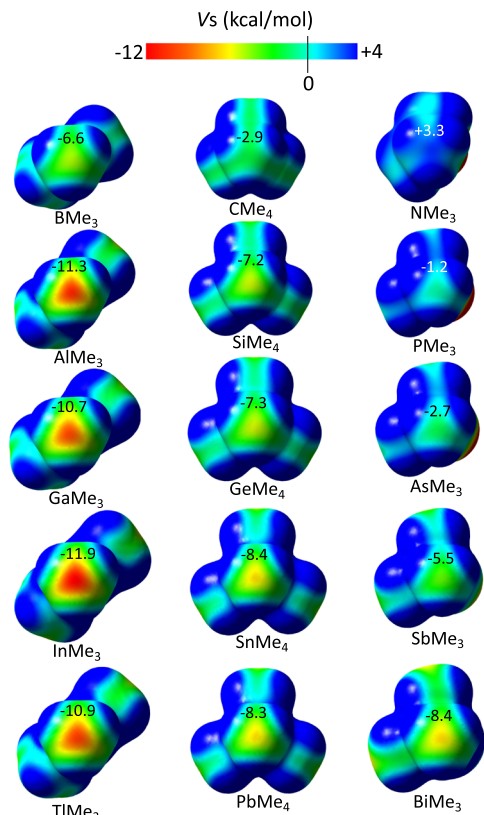

**Fig. 1 Electron density distribution for differently substituted methyl groups.** Molecular electrostatic potential (MEP) maps for compounds E(CH₃)₃ (E = B, Al, Ga, In, Tl), E′(CH₃)₄ (E′ = C, Si, Ge, Sn, Pb), and E″(CH₃)₃ (E″ = N, P, As, Sb, Bi) with the minimum value of the EP ($V_{s,min}$) at the center of the tetrahedral CH₃ face given in kcal mol⁻¹.

electronegative than carbon) would lead to the opposite situation, that is, a negatively charged carbon atom. We have plotted the MEP for differently substituted methyl groups in simple neutral species and the corresponding pictures are shown in Fig. 1. As expected, carbon atoms connected to electropositive atoms, for example in AlMe₃, show negative electrostatic potential ($V_s$) values, while those bound to more electronegative atoms, like in NMe₃, show positive EP. Remarkably, for all groups 13, 14, and 15 (with the exception of N) compounds, the methyl group exhibits a carbon atom with negative $V_s$. Therefore, the conditions needed to have methyl groups that can electrostatically interact, via its C atom, with a Lewis acid do not seem particularly restrictive. It is also noteworthy that there is a nice dependence of the $V_s$ with the Pauling electronegativity of the substituent atom (Fig. 2) that follows the period of the periodic table rather than the group.

**Topology of the electron density and geometrical features of experimental examples**. To check whether these interactions are just hypothetical or really existing, we have searched the Cambridge Structural Database (CSD)[34] for E-CH₃⋯Y short contacts in which E is an atom less electronegative than carbon (E = B, Al, Ga, In, Tl, Si, Ge, Sn, Pb, P, As, Sb, and Bi) and Y is a Lewis acid. Remarkably, these short contacts are in fact ubiquitous and many examples have been found for a plethora of different Lewis acids. To complement the structural study, we have analyzed the topology of the electron density of some selected experimental examples. Although the Quantum Theory of Atoms in Molecules[35] (QTAIM) has been widely used for the analysis of

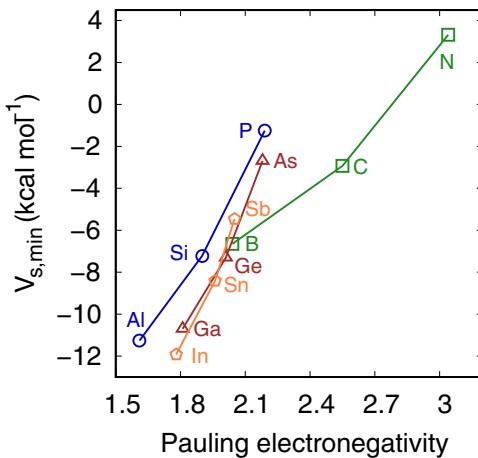

**Fig. 2 The role of the electronegativity in the charge of the carbon atom.** $V_{s,min}$ value at the center of the tetrahedral $CH_3$ face as a function of the Pauling electronegativity of the central atom for compounds $E(CH_3)_3$ (E = B, Al, Ga, In, Tl), $E'(CH_3)_4$ (E' = C, Si, Ge, Sn, Pb), and $E''(CH_3)_3$ (E'' = N, P, As, Sb, Bi).

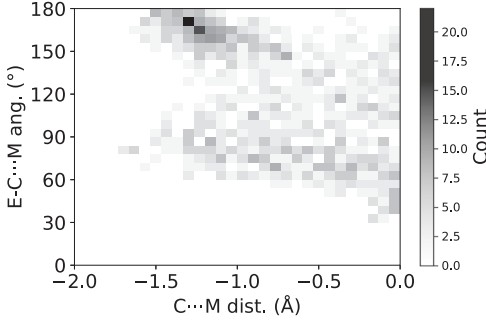

**Fig. 3 Geometrical trends in alkaline metal bonding.** Angles (E-C···M) distribution as a function of the C···M vdW-normalized distance for M = Li (black), Na (green), K (blue), and Rb (red). E can be any group 13, 14 (except C), and 15 (except N) atom. Source data are provided as a Source Data file.

noncovalent interactions, there is an ongoing debate about the validity of bond paths as evidence of attractive interactions[36,37]. Recently, Jablonski has demonstrated the limitations of QTAIM to determine dominant interactions between distant atoms[38,39]. Alternatively, noncovalent interactions[40] (NCI) method partitions real space in different bonding regions depending on the values of the electron density and its reduced density gradient. These regions allow clearly distinction between strong attractive (blue), weak noncovalent (green), and repulsive (red) interactions. We have used here a combination of both methods, QTAIM and NCI, to get a more general picture of the topological features of the electron density within the interaction region.

Since one of the most important features of noncovalent interactions based on electrostatic forces is their directionality, in search for possible alkaline metal bonding, the E-C···M angle distribution as a function of the C···M distance is plotted in Fig. 3 (M = Li, Na, K, Rb, Cs). Two main trends can be observed: there is a first group of structures in which the angle approaches 80–90° as the C···M distance shortens. This angle value corresponds to the extension of one of the C-H bonds, a region that, despite not being that of most negative EP, still has a charge accumulation according to the MEP maps of Fig. 1. Moreover, this peak is also consistent with the presence of secondary interactions coexisting with the carbon···alkaline short contact as, for instance, in the

crystal structure of RIFHAP[41], in which the potassium cation clearly interacts with a $sp^3$-nitrogen atom at a short distance ($d_{K···N}$ = 2.431 Å). On a second group, the E-C···M moiety becomes more linear (E-C···M angles close to 180°) as the C···M distance makes shorter. This interaction geometry, which should maximize the electrostatic attraction since it involves the most negative region ($V_{s,min}$) of the methyl group, concentrates the majority of crystal structures in Fig. 3.

We have observed that in methyl···alkaline metal contacts, the alkaline cation can be both naked and coordinated. In Fig. 4, we present an example of both cases. In the crystal structure of GUDGAO[42], a dicoordinate Li center acts as the Lewis acid, while in the crystal structure of CUVMAH[43] this role is played by a naked $Na^+$ cation (Fig. 4a, b, respectively). In both cases, the direct carbon···metal interaction (vdW-corrected C···M distances for GUDGAO and CUVMAH are –1.367 and –1.065 Å, respectively) is confirmed by the presence of bond paths when inspecting the electron density by means of QTAIM calculations. Similarly, methyl···alkaline-earth metal short contacts can also be found in the CSD. A nice example is given in the crystal structure of JEJGIO[44], in which the very short methyl···magnesium contact (vdW-corrected C···Mg distance = −1.745 Å) has associated a bond path in the QTAIM graph (Fig. 4c). Similar interactions with Be, involving more typical electron density donors containing lone pairs, but with the same interaction topology, were described by Yáñez et al.[45]. Remarkably, the NCI surfaces show a high directionality of the interactions in the three examples. It is worth mentioning that, although some alkaline(-earth)···methyl contacts have been previously identified in different crystal structures, they have been traditionally considered as agostic interactions rather than as structure-driving noncovalent bonds[46].

We have also investigated the existence of methyl groups as triel bonding acceptors. It was shown in a previous work the poor capability of boron to be involved in this type of interactions[22], which is in good agreement with the less negative calculated EP with respect to the other group 13 elements (see Fig. 1) and with the scarce number of boron structures found at short interacting distances. CSD searches have unveiled a marked tendency for linear E-C···Y (Y being a group 13 atom except B) moieties as the C···Y distance shortens (Fig. 5a). For planar trisubstituted group 13 compounds, the C···Y-R angle, which indicates the position of the methyl group with respect of the triel molecular plane, mostly tends to 90° for short distances, although other interaction topologies are found in the case of non-planar molecules (Fig. 5b). This is a strong indication of triel bonding since such interaction geometry maximizes the electrostatic attraction by connecting the $V_{s,min}$ of the methyl with the $V_{s,max}$ of the triel center. The crystal structure of trimethylgallium (OFURUC)[47], with two molecules connected via a carbon···gallium bond path (Fig. 5c; vdW-corrected C···Ga distance = –0.941 Å), nicely illustrates the π-hole interaction with a methyl group behaving as electron density donor. The planar nature of trimethylgallium allows the establishment of H···C and H···H attractive interactions[48] accompanying the C···Ga short contact as disclosed by the computed NCI isosurfaces (Fig. 5c).

Following the groups of the periodic table, tetrel, pnictogen, chalcogen, and halogen bonds are σ-hole-based interactions and, accordingly, one could imagine methyl groups participating in them as Lewis bases, in a similar fashion to what we have seen for alkali metal, alkaline-earth metal, and π-hole bonding. Remarkably, we have found several structures in which methyl groups act as electron density donors in tetrel, pnictogen, chalcogen, and halogen interactions. An interesting example of tetrel bonding is found between a trisubstituted Pb(II) center and a methyl group in the crystal structure of KAPTEB[49]. There, the short Pb···C

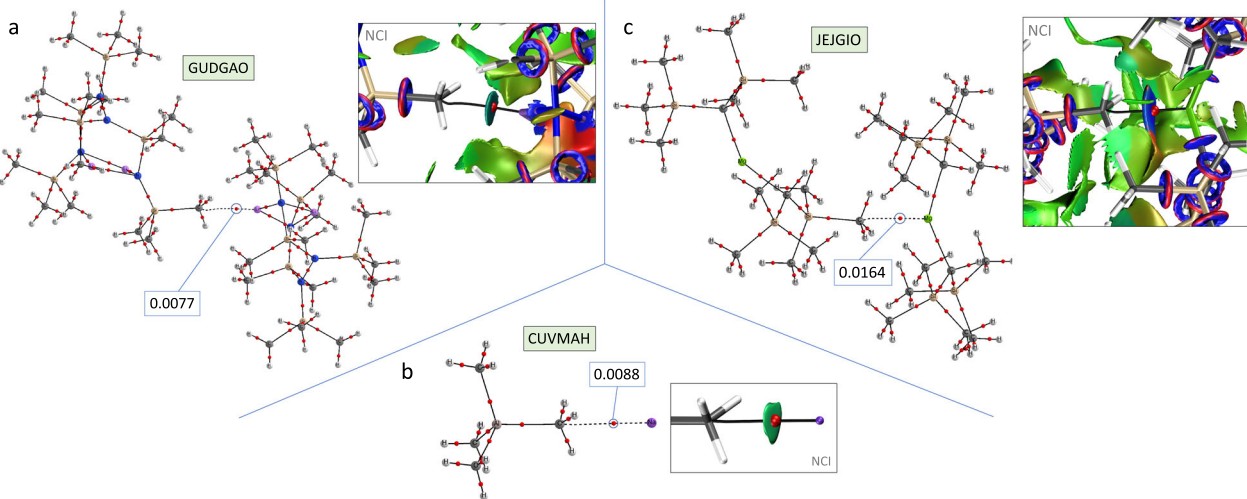

**Fig. 4 Alkaline and alkaline earth metal bonding.** QTAIM and NCI graphs showing the methyl···alkaline(-earth) metal interaction in the crystal structures of **a** GUDGAO, **b** CUVMAH, and **c** JEJGIO. The values of the electron density at the bond critical point (BCPs) are given in atomic units (a. u.).

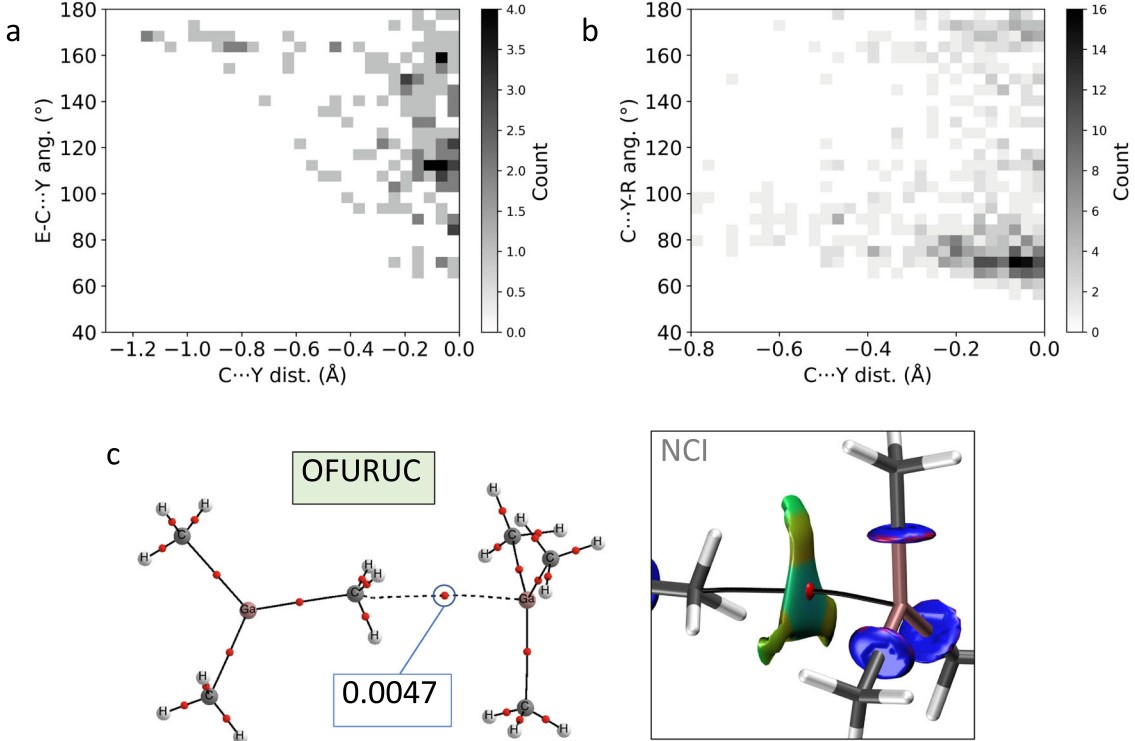

**Fig. 5 Triel bonding. a** E-C···Y and **b** C···Y-R angles distribution as a function of the C···Y vdW-normalized distance for Y = Al, Ga, In, and Tl. E can be any group 13, 14 (except C), and 15 (except N) atom. Source data are provided as a Source Data file. **c** QTAIM and NCI graphs showing the methyl···gallium interaction in the crystal structure of OFURUC. The values of the electron density at the BCPs are given in a. u.

contact at 3.84 Å is associated with a bond path in the QTAIM analysis (Fig. 6a). For group 15 elements, we show pnictogen interactions involving Sb atoms and methyl groups in the crystal structure of OMECUF[50] (see the molecular structure and bond path in Fig. 6b). We also show an interesting example of chalcogen bonding (ZUNXIS[51], Fig. 6c) between a methyl group and a selenium atom. Note the practically linear orientation of the methyl group relative to the C-Se covalent bond that is characteristic of chalcogen bonds. Despite not being included in the heat maps, these two last examples (OMECUF and ZUNXIS) serve to illustrate the fact that methyl groups can also behave as Lewis bases when connected to another carbon atom since,

according to Fig. 1, the value of the MEP is still slightly negative at the center of the methyl tetrahedral face.

There is also a considerable number of methyl···halogen interactions at distances shorter than the sum of the corresponding vdW radii. An example of halogen bonding between a methyl group and a bromine atom is presented in Fig. 7a. It is interesting to notice in that crystal structure (XUVSEN[52]) the linear arrangement of the C-C···Br-C framework that is a feature of halogen bonds. The high directionality of the interaction can be seen in the NCI plot in Fig. 7a. In fact, it is a general trend that both the E-C···X and C···X-C angles tend to 180° as the C···X interatomic distance decreases, as depicted in Fig. 7b, c, which

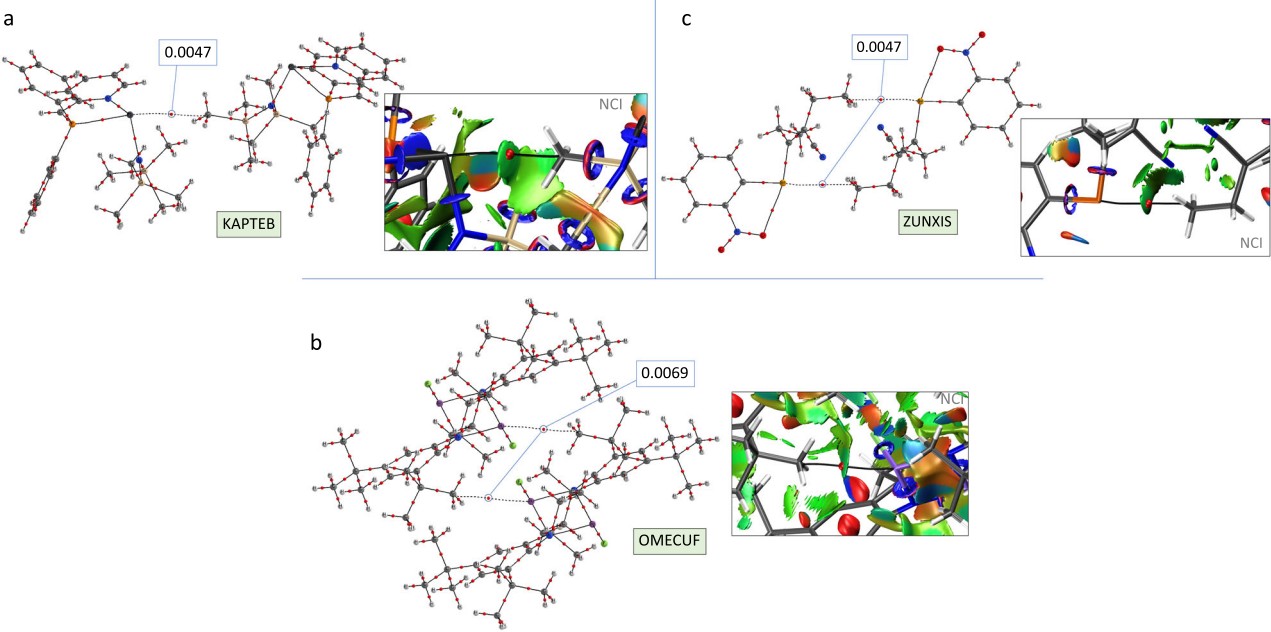

**Fig. 6 Tetrel bonding.** QTAIM and NCI graphs showing the methyl···Y interaction in the crystal structures of **a** KAPTEB, **b** OMEFUC, and **c** ZUNXIS (Y = Pb, Sb, and Se, respectively). The values of the electron density at the BCPs are given in a. u.

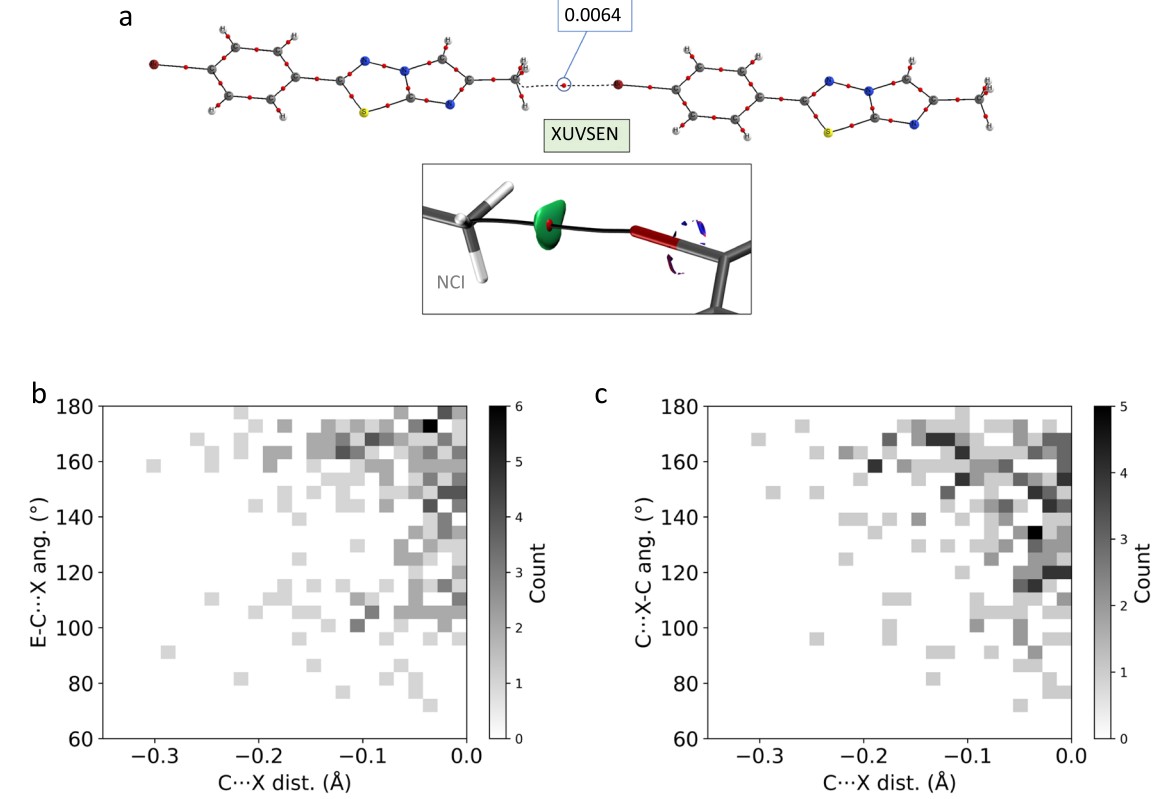

**Fig. 7 Halogen bonding. a** QTAIM and NCI graph showing the methyl···bromine interaction in the crystal structure of XUVSEN. The values of the electron density at the BCPs are given in a. u. **b** E-C···X and **c** C···X-C angles distribution as a function of the C···X vdW-normalized distance for X = Br and I. E can be any group 13, 14 (except C), and 15 (except N) atom. Source data are provided as a Source Data file.

reinforces the evidence of the existence of methyl groups as halogen bonding acceptors. In some cases, the E-C···X angle shows a tendency toward values close to 80–100° as a consequence of secondary interactions between the halogen atom and other nucleophilic regions of the molecules. For instance, in XUVSEN (Fig. 7a), a hypothetical interaction of the bromine with the nitrogen atom of the imidazole ring would involve a C-C···Br angle of ca. 80°.

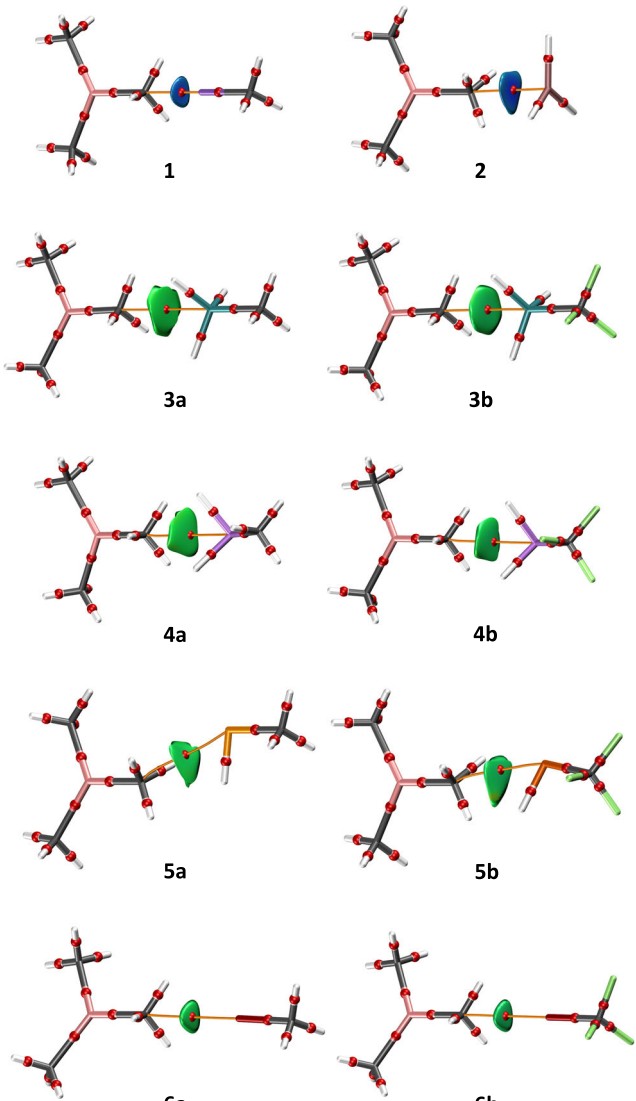

**Fig. 8 Model systems.** NCI graphs of the optimized adducts **1**, **2**, **3a**–**6a**, **3b**–**6b** formed by Al(CH$_3$)$_3$ and several Lewis acids.

**Table 1 M06-2X/def2-TZVPD interaction energies, absolute ($d_{C\cdots Y}$), and vdW-corrected ($d_{vdW}$) interatomic distances and Al-C⋯Y and C⋯Y-C angles for optimized model adducts 1–6.**

| System | $d_{C\cdots Y}$ (Å) | $d_{vdW}$ (Å) | Al-C⋯Y ang. (°) | C⋯Y-C ang. (°) | $\Delta E_{BSSE}$ (kcal mol$^{-1}$) |
|---|---|---|---|---|---|
| **1** | 2.27 | −1.62 | 178.6 | 177.9 | −9.43 |
| **2** | 2.73 | −1.36 | 178.0 | 93.7, 94.0, 94.1[a] | −8.48 |
| **3a** | 3.61 | −0.45 | 178.0 | 179.6 | −1.25 |
| **3b** | 3.45 | −0.61 | 178.8 | 179.6 | −2.22 |
| **4a** | 3.59 | −0.06 | 178.8 | 156.8 | −1.17 |
| **4b** | 3.46 | −0.19 | 178.6 | 156.1 | −2.05 |
| **5a** | 3.57 | −0.02 | 155.7 | 156.0 | −1.24 |
| **5b** | 3.42 | −0.17 | 171.1 | 160.6 | −1.93 |
| **6a** | 3.47 | −0.16 | 178.0 | 179.6 | −0.85 |
| **6b** | 3.36 | −0.27 | 179.0 | 179.8 | −1.58 |

[a]In GaH$_3$ the angle values refer to the three C⋯Ga-H angles.

character, in good agreement with the higher calculated interaction energies for those systems. In the other adducts, the attractive interaction is of noncovalent nature (green surfaces). In **6a** and **6b**, the bonding region is small and reduced to a C⋯Br interaction, whereas in **3a**, **3b**, **4a**, and **4b**, the bonding surface becomes larger, showing incipient interactions between the hydrogen atoms of the methyl group and the Lewis acid. It is worth commenting that in **5a** and **5b**, according to the position of the σ-hole of the selenium atoms (see MEP maps in Supplementary Fig. S1), the Al-C⋯Se-C framework significantly deviates from linearity. By looking at the NCI surfaces of the optimized structures, it seems that C⋯H hydrogen bonds are established in **5a** and **5b** coexisting with the C⋯Se chalcogen bonds. Remarkably, inspection of the QTAIM electron density confirms the presence of C⋯Se bond paths both in **5a** and **5b**.

**Natural bond orbital analysis**. It is well known that charge transfer processes based on orbital overlap play an important role in many noncovalent interactions, particularly in those with interaction distances much shorter than the sum of the corresponding vdW radii. In consequence, we have performed a natural bond orbital (NBO) analysis of systems **1**, **2**, **3b**–**6b** to shed light into the nature of methyl⋯Lewis acid interactions (see complete results in Supplementary Table S2). In general, we have observed that orbital interactions are the combination of charge transfer processes from occupied orbitals of the methyl group (σ$_{C-H}$ and σ$_{C-Al}$) into empty antibonding orbitals of the electrophile. Those acceptor orbitals are a p orbital of Ga in the case of **2**, and σ*$_{C-E}$ orbitals for **3b**, **4b**, **5b**, and **6b** (E = Ge, As, Se, and Br, respectively). Remarkably, these orbitals interactions are clearly associated with π- and σ-hole bonds. On the other hand, in **1**, although the donor orbitals are the same as in the aforementioned cases, the acceptors are empty s orbitals of Li. Further NBO calculations in CUVMAH, in which the alkaline metal is a naked cation (see Fig. 4), have shown that the orbital interaction is similarly dominated by a σ$_{C-Al}$ → s*$_{Na}$ charge transfer.

**Shape analysis**. By looking at the short interatomic distances shown in Table 1 (particularly in **1** and **2**), a distortion on the geometry of the interacting methyl groups could be expected. To try to shed light into this, we have performed a continuous shape measure (CShM)[53,54] analysis of the methyl groups in those model systems. CShM is a powerful tool that has been extensively

**Model systems**. To understand the nature and strength of the interactions involving methyl groups as electron density donors, we have investigated several model adducts with trimethyl-aluminum as the Lewis base and different Lewis acids, namely LiCH$_3$ (**1**), GaH$_3$ (**2**), GeH$_3$CH$_3$ (**3a**), GeH$_3$CF$_3$ (**3b**), AsH$_2$CH$_3$ (**4a**), AsH$_2$CF$_3$ (**4b**), SeHCH$_3$ (**5a**), SeHCF$_3$ (**5b**), BrCH$_3$ (**6a**), and BrCF$_3$ (**6a**) (Fig. 8). The main geometrical features of the optimized systems along with the calculated interaction energies are shown in Table 1. A few observations can be made: (1) in general, all geometries are in good agreement with those expected for each type of noncovalent interaction based on the MEP maps of the molecules involved (see Supplementary Fig. S1); (2) all interactions are attractive and Al(CH$_3$)$_3$⋯LiCH$_3$ (**1**) and Al(CH$_3$)$_3$⋯GaH$_3$ (**2**) adducts feature the strongest interactions (−9.43 and −8.48 kcal mol$^{-1}$, respectively); and (3) replacing the methyl groups attached to the atom acting as Lewis acid with an electron-withdrawing group as CF$_3$ reduces the intermolecular distances and reinforces the interaction strength by 70–86%. All these results support the hypothesis that methyl groups can act as Lewis bases in noncovalent interactions.

The NCI analysis of the model systems discloses significantly directional interactions (Fig. 8). In **1** and **2**, the blue isosurfaces indicate a strong interaction with some degree of covalent

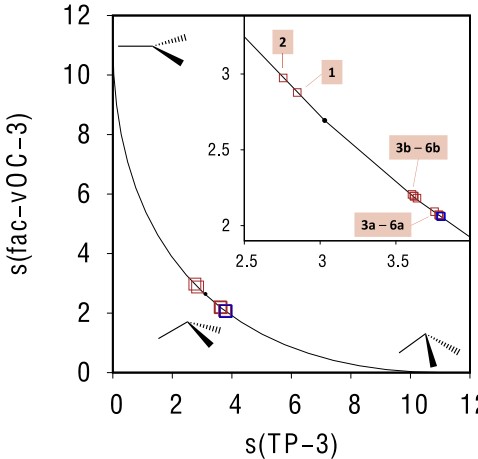

**Fig. 9 Shape analysis of the interacting methyl group.** Minimal Interconversion pathway between a fac-vacant octahedron (fac-vOC-3) and a planar trigonal (TP-3) shape with the corresponding continuous shape measure (s) values (red squares) for the interacting methyl group of Al(CH₃)₃ in all model systems (**1**, **2**, **3a–6a**, **3b–6b**). The blue square, which represents the methyl group of a non-interacting Al(CH₃)₃ molecule, is shown as reference. The small black dot refers to the position in the curve of a vacant tetrahedron.

used to calibrate the degree of distortion of a chemical system with respect to a reference polyhedral shape[55–57]. In the context of CShM, a value of shape measure $s = 0$ indicates that the measured structure has exactly the same shape as the reference polyhedron, whereas increasing values of s are associated with more distorted shapes. Furthermore, minimal interconversion pathways (MIPs) allow the identification of structures with intermediate geometries between two reference polyhedra with the same number of vertex[58]. In Fig. 9, we represent the MIP between a planar trigonal (TP-3) structure and a vacant octahedron (fac-vOC-3), with the tetrahedron represented as a small black dot over the MIP curve. In this way, we can calibrate the degree of pyramidalization of the -CH₃ group, which is usually indicative of an interaction with non-negligible orbital character.

Remarkably, if we represent the corresponding CShM values (s) for the interacting methyl groups of **1**, **2**, **3a–6a**, **3b–6b** (and also the methyl group of AlMe₃ in the molecule alone, i.e., not interacting, as a reference) with respect to TP-3 and fac-vOC-3, three different groups of points appear: a first group, comprising **3a–6a**, very close to the undistorted reference shape (blue square); a second group with **3b–6b**, still close to the reference shape but distorted toward a more planar CH₃ arrangement (i.e., closer to TP-3); and finally a third group with **1** and **2**, which is clearly more planar and significantly distant from the original undistorted CH₃ geometry. These groups with different degrees of geometry changes are in very good agreement with the differences observed in the interaction strengths and the corrected interatomic vdW distances. We believe that the fact that significant geometrical rearrangements are observed in the methyl groups upon interaction reinforces the main proposition of this article.

In summary, we have shown herein that methyl groups, when connected to elements (E) that are less electronegative than carbon, can play the role of nucleophiles in noncovalent interactions. By computing MEPs of the molecules involved, we have demonstrated that the physical origin of these interactions is electrostatic. NBO calculations have also unveiled charge transfer contributions, with bonding σ$_{C-E}$ and σ$_{C-H}$ orbitals acting as the electron density donors. Remarkably, there are hundreds of experimental examples of these methyl···Lewis acid interactions,

the Lewis acid being an alkaline, alkaline earth, triel, tetrel, pnictogen, chalcogen, or halogen atom. The significant directionality of the interactions along with their strength evidences that they are structure-driving and open the door to a new supramolecular chemistry based on the capability of electron-rich methyl groups to behave as Lewis bases.

## Methods

**Structural searches.** Searches were done in the CSD[34] version 5.41 (November 2019) + 3 updates. Only structures with 3D coordinates defined, non-disordered, with no errors and with $R < 0.1$ were allowed in searches. CSD identifiers are given as six-letter refcodes throughout the text (e.g., ABCDEF). In all searches regarding E-CH₃···Y short contacts, E was restricted to any group 13, 14 (except C), and 15 (except N) atom, whereas Y was modified depending on the periodic group under study. Only interatomic C···Y distances shorter than the sum of the van der Waals radii were included, with no further constraint. We used the van der Waals radii proposed by Alvarez[59].

**Electronic structure calculations.** DFT calculations were carried out with Gaussian16[60] at the M06-2X level and with the def2-TZVPD basis sets for all atoms, with the corresponding pseudopotentials for atoms of period 5 of the periodic table. The M06-2X functional has shown a good performance for the study of noncovalent interactions in previous benchmark analysis[61,62]. All interaction energies were calculated via the supermolecule approach ($\Delta E_{AB} = E_{AB} - E_A - E_B$) and corrected for the basis sets superposition error via the counterpoise method[63]. All optimized systems were characterized as minima of the corresponding potential energy surfaces by inspection of the eigenvalues of the diagonalized Hessian matrices. Geometries from crystal structures retrieved from the CSD were kept fixed at their crystallographic coordinates to simulate specific interaction topologies present in the solid state. MEP maps were built on the 0.001 a. u. isosurface of the electron density with GaussView[64]. NBO analyses were done with the NBO3.1 software[65] as implemented in Gaussian16 at the M06-2X/def2-TZVP level.

**Topological analysis of the electron density.** QTAIM analyses were done on the DFT wavefunctions with AIMAll[66]. Only the bond paths between the donor methyl and the acceptor atoms are depicted in the QTAIM graphs for the sake of clarity. NCI calculations[40] were performed on promolecular densities with MultiWfn 3.7[67] and the corresponding isosurfaces (s = 0.3 a. u.) were represented with VMD 1.9.3[68]. The surfaces are colored according to values of sign($\lambda_2$)$\rho$, blue indicates strong attractive interactions, green indicates weak noncovalent attraction, and red indicates strong nonbonded overlap. CShM analysis was carried out with the software SHAPE 2.1[69].

## Data availability

The data that support the findings of this article are available from the corresponding author upon reasonable request. A Source Data file is provided with this paper.

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

## Acknowledgements

J. E. thanks the Spanish MICCIN for a *Ramón y Cajal* research contract (RYC-2017-22853). Financial support from the Spanish Structures of Excellence *María de Maeztu* program (MDM-2017-0767) and Spanish MICINN through grant PID2019-109119GA-I00 are gratefully acknowledged.

## Author contributions

O. L. performed the electronic structure calculations. J. E. conceived the project, carried out the structural analysis, and wrote the article. Both authors revised the article and approved the final version for publication.

## Competing interests

The authors declare no competing interests.
