## [Peer Review File · Nature Communications]

Methyl groups as widespread Lewis bases in noncovalent interactionsREVIEWER COMMENTS

Reviewer #1 (Remarks to the Author):

As the authors explain in their introduction, there is an extensive list of noncovalent bonds that have been established and explored in recent years. Most of the prior focus has been on the atoms and groups of the Lewis acid entity from which each bond derives its name. Regarding the base, electron donation has been considered primarily from lone pairs and pi systems. However, there have been some exceptions where the density is extracted from sigma bonding orbitals. Taking their cue from some recent work, the authors reason that since C is reasonably electronegative, bonding it to an electropositive atom ought to pile up some density on the C, providing it with the opposite of a sigma hole, which would acquire some negative rather than positive charge. The calculations in the first part of this paper verify this presumption, documenting negatively charged regions near the C atoms of methyl groups.

The next segment of the manuscript surveys the relevant crystal databases to look for structures which are consistent with their hypothesis. The figures do support their contention, with the proviso that these bonds are probably not very strong so the geometrical trends are not fully compelling, more in the line of indicative. Quantum calculations on several sample structures further buttress the idea, with bond paths including the pertinent C atom.

The last component of the paper systematically develops a number of model dimers, which are then fully optimized, after which the authors examine their bonding characteristics, such as AIM and NBO, and including the all-important interaction energies which enable them to place these interactions on the overall scale of noncovalent bonds. The energetics are weak but respectable, large enough that the point has been made.

All in all, the authors have developed their ideas, and then furnished solid evidence, which include quantitative values. While there has been some earlier activity in examining crystals for evidence of C-tetrel bonds, this work has considered C as the electron acceptor. The idea that C can act as donor has enjoyed little attention to this point. Publication of this paper in Nat. Comm. will advance the field by putting this idea out there to a wide audience, some of whom may be inspired to search their own structures for bonds of this type. So publication is recommended with two minor suggestions below.

1- A somewhat better rationale should be provided for the set of points in Fig 3 that approach 90 degs.

2- It is unclear what is meant by "vdW corrected distances" that have negative values. It might be better to use instead the ratio between the interatomic distance and the sum of atomic vdW radii.

Reviewer #2 (Remarks to the Author):

Report and Supplementary file attached

Reviewer #3 (Remarks to the Author):

In the article, the authors try to convince the reader that the methyl group, through its face, can quite often participate in intermolecular interactions with almost any Lewis acid.

This idea is interesting and the method of conducting research is appropriate.

However, it seems to me that the authors should be a bit more reserved in this respect and try to defend their thesis more. Although the authors support this idea with 'evidence' from the crystallographic database, it does not convince me entirely.

The doubts are as follows:

1) Figures 3, 5a, and 7b show in fact a considerable angular spread even outside the range of ca. 80-100 degrees.

2) The authors clearly support their idea with the presence of BP, which, however, is not as clear as it might seem. Whether or not BPs should be believed so much has been extensively debated in the literature and has been perfectly summarized in recent articles by Jablonski, Shahbazian, and Clark,

see for example [J. Comput. Chem. 2018 39 2183; ChemistryOpen 2019 8 497; Chem. Eur. J. 2018 24 5401; J. Mol. Model. 2018 24 142] The authors should mention this at least briefly.

3) Distances C...Y are rather long.

4) Instead of talking directly about the C...Y interaction (how to prove its existence?), it is probably better to refer to Me...Y. Then, however, the correct alignment of Me can be considered as a 'geometric combination' of three sigma bond...Y interactions.

Therefore, in this matter, I give priority to the theoretical model systems considered by the authors in the last part of the article. Although this part is probably more convincing, I feel unsatisfied with the techniques used:

5) QTAIM is troublesome (see comment above)

6) The result from NBO is rather obvious as the NBO method is structured to give MOocc-> MOvirt interactions. The methyl group is closest, so what orbitals other than sigma_CH and sigma_AIC were supposed to interact?

7) I think that the dimers investigated are perfect to use the NCI method and I am surprised that this has not been done. The NCI method should elegantly show whether the electron density is mainly located on the C...Y axis or within H...Y. NCI is available in the AIMAll used.

8) The authors could think of still other methods showing electron density localization (ELF, Laplacian or L, etc.), although some may require another programs.

Nevertheless, in summary, I believe that the present article deals with an interesting issue and may cause a broader debate among chemists, so it should be published. However, before that, the authors should consider the above remarks (especially describe the allegations regarding QTAIM) and supplement the theoretical analysis with, at least, the NCI calculations.

As the correction requires additional calculations, my grade is MAJOR corrections.

Other (minor) comments go only to the authors.

Minor:

2/7 & 12/6: should be "occupied orbital"

2/13: should be "Lewis acids"

2/22: "for instance" should be removed because, according to the accepted nomenclature, 'acidic' C always forms tetrel bonds

2/23: Carbon atoms cannot be "in the form of carbenes".

2/26: "... density donors." - references should be added

2/31: "we can ask ourselves"; stylistically awkward, maybe, e.g. 'we may ask if ...'

3/7: molecular electrostatic potential (MEP)

3/12: "leading to"; better would be ", that is ..."

3/6&13: The use of the expressions "tetrasubstituted" and "substituted" is confusing because it is difficult to say that in the group -CH₃ a carbon atom is substituted, especially with four substituents.

7: The plot for the C...Y-R angle should also be shown because 5a alone is not too convincing.

11: Dimer 2 is missing in Figure 8.

11/3&7: should be "selenium atoms" and "C...Se bond paths", respectively

Table 1: The authors should rather report absolute distance than vdW-corrected ones, especially in the theoretical part (Table 1).

12/17: "demonstrated...physical origin ... is electrostatic" - The authors should clearly state how they think

they have found this electrostatic origin.

13/2: "The high directionality" - this seems to be somewhat exaggerated

Dr. Miroslaw Jablonski

Reviewer #2 Attachment:

In this article, the authors highlight the capacity of a methyl C-atom to function as the electron rich partner in intermolecular interactions with Lewis acidic sites and postulate that this is a rather general phenomenon within crystal structures. These assertions are further undergirded by computational studies, such as the electrostatic potential maps and the geometry optimization of several model bimolecular adducts. The study is adequately conducted and the conclusions are consistent.

While the idea is good and the message of the generality of the concept can be seen as new, the authors should have contextualized the narrative better because many others have previously noted the ability of C to act as electron rich site in a noncovalent interaction. I also have some other general queries and provided that the authors can quell my concerns the paper can become suitable to be published in *Nature Communications*.

1. Better contextualization of the idea

The idea conveyed in the paper is not really new and the authors should (have) referred better to the existing literature. One article from the author himself already reported on the ability of methyl groups to act as electron rich sites for B, Al, Ga, and In: <https://doi.org/10.1039/C7CE01259D> (this one is cited as reference 15). The short contact they highlight in structure JEJGIO was already noted (even in the title) the C --- Mg contact (<https://doi.org/10.1039/C39900000847>). Moreover, papers where a methyl group has been noticed to act as electron rich entity date back to the early 1990ies (mostly studies of interactions with water), none of which was cited in the article:

<https://doi.org/10.1063/1.465050>;
<https://doi.org/10.1063/1.466569>;
<https://doi.org/10.1063/1.468280>;
<https://doi.org/10.1016/j.cplett.2008.11.009>;
<https://pubs.acs.org/doi/abs/10.1021/jp202704c>;
<https://pubs.acs.org/doi/abs/10.1021/ct300132e>;
<https://doi.org/10.1039/C5CP03704B>;
<https://doi.org/10.1007/s12039-015-0861-7>;
<https://doi.org/10.1134/S0036024416100150>;
<https://doi.org/10.1016/j.comptc.2016.06.014>;
<https://doi.org/10.1002/cphc.202000927>,
<https://doi.org/10.3390/molecules24183370>,

2. Additional literature references

There are several recent important papers dealing specifically with interactions involving methyl groups that should be cited:

<https://doi.org/10.3390/molecules24183370>;
<https://doi.org/10.3390/c6040060>;
<https://doi.org/10.1002/anie.201811171>

The reference to tetrel bonding interactions (ref 10) misses the most relevant papers

<https://doi.org/10.1039/C3CP53369G>;
<https://doi.org/10.1002/anie.201306501>;
<https://doi.org/10.1002/tcr.201500256>;

3. Tetrahedral distortions

The reported van der Waals corrected distances with the alkali metals are very small and reported to be up to -1.6 \AA . This would imply some covalent character and as a result I would expect that the tetrahedral

geometry of a methyl carbon will be distorted. However, this is not reflected in the data: there is no correlation between the van der Waals corrected distance and the shortest E–C–H angle. Do the authors have an explanation for this? I suspect this is also the case for the other datasets.

4. A redo of a CSD search gave different data

In Figure 3 of the paper, the authors present a CSD search for examples of E–CH₃···X structures where the C···X distances are shorter than the van der Waals radii according to Alvarez. They plot the ECX angle as a function of the van der Waals corrected CX distance, revealing that the densest feature is at very short distances and near an ECX angle of 180 degrees. This would be consistent with their idea that methyl-C is a directional ‘sigma-hole acceptor’. Intrigued by these data, I wanted to repeat the search but came to very different data when using some looser criteria and when applying the criteria described in the paper. In the data I obtained, the feature near 180 degrees is the minor feature by far. Can the authors explain this discrepancy? The conquest file is included for the authors and the plots are given below.

5. Minor issues

- The definition of E is different in Fig.1 and 2 vs 3. This has to be fixed.
- Page 2, line 17: '...other electron-rich species had...' should be 'have'.
- The use of personalized language should be modified (e.g. 'we first need to', 'let us look', etc.).
- Page 3, MEP should be defined as 'molecular electrostatic potential' and EP should be defined.
- Why is the CSD data not plotted as heat plots? I think those are much clearer and more informative because the most dense regions are immediately obvious.
- The term 'nucleophile' is typically used to describe the attacking entity in a chemical *reaction* and some might thus take issue with its use in the context of an *interaction*. My advice is to avoid the ter

REVIEWER COMMENTS

Reviewer #1 (Remarks to the Author):

As the authors explain in their introduction, there is an extensive list of noncovalent bonds that have been established and explored in recent years. Most of the prior focus has been on the atoms and groups of the Lewis acid entity from which each bond derives its name. Regarding the base, electron donation has been considered primarily from lone pairs and pi systems. However, there have been some exceptions where the density is extracted from sigma bonding orbitals. Taking their cue from some recent work, the authors reason that since C is reasonably electronegative, bonding it to an electropositive atom ought to pile up some density on the C, providing it with the opposite of a sigma hole, which would acquire some negative rather than positive charge. The calculations in the first part of this paper verify this presumption, documenting negatively charged regions near the C atoms of methyl groups.

The next segment of the manuscript surveys the relevant crystal databases to look for structures which are consistent with their hypothesis. The figures do support their contention, with the proviso that these bonds are probably not very strong so the geometrical trends are not fully compelling, more in the line of indicative. Quantum calculations on several sample structures further buttress the idea, with bond paths including the pertinent C atom.

The last component of the paper systematically develops a number of model dimers, which are then fully optimized, after which the authors examine their bonding characteristics, such as AIM and NBO, and including the all-important interaction energies which enable them to place these interactions on the overall scale of noncovalent bonds. The energetics are weak but respectable, large enough that the point has been made.

All in all, the authors have developed their ideas, and then furnished solid evidence, which include quantitative values. While there has been some earlier activity in examining crystals for evidence of C-tetrel bonds, this work has considered C as the electron acceptor. The idea that C can act as donor has enjoyed little attention to this point. Publication of this paper in Nat. Comm. will advance the field by putting this idea out there to a wide audience, some of whom may be inspired to search their own structures for bonds of this type. So publication is recommended with two minor suggestions below.

We warmly thank the referee for the kind comments. We also feel that these are some interesting results and with the additions and modifications made we hope the manuscript is now suitable for publication in Nat. Commun.

1- A somewhat better rationale should be provided for the set of points in Fig 3 that approach 90 degs.

Fig. 3 and the other figures related to the structural analysis have been redone as 2D histograms to be clearer. Furthermore, we have further discussed on the presence of the peak at around 80-90°.

2- It is unclear what is meant by "vdW corrected distances" that have negative values. It might be better to use instead the ratio between the interatomic distance and the sum of atomic vdW radii.

We understand the referee's concern, but we prefer to keep this nomenclature since it has been extensively used by us in the past and we believe that it is an easy way to immediately see how far the contact distance is from the van der Waals radii sum.

Reviewer #2 (Remarks to the Author):

In this article, the authors highlight the capacity of a methyl C-atom to function as the electron rich partner in intermolecular interactions with Lewis acidic sites and postulate that this is a rather general phenomenon within crystal structures. These assertions are further undergirded by computational studies, such as the electrostatic potential maps and the geometry optimization of several model bimolecular adducts. The study is adequately conducted and the conclusions are consistent.

While the idea is good and the message of the generality of the concept can be seen as new, the authors should have contextualized the narrative better because many others have previously noted the ability of C to act as electron rich site in a noncovalent interaction. I also have some other general queries and provided that the authors can quell my concerns the paper can become suitable to be published in Nature Communications.

We appreciate the referee's comments on our work and the effort in reading and making suggestions to improve the manuscript. In the next lines, we will try to address all her/his concerns.

1. Better contextualization of the idea

The idea conveyed in the paper is not really new and the authors should (have) referred better to the existing literature. One article from the author himself already reported on the ability of methyl groups to act as electron rich sites for B, Al, Ga, and In: <https://doi.org/10.1039/C7CE01259D> (this one is cited as reference 15). The short contact they highlight in structure JEJGIO was already noted (even in the title) the C - - Mg contact (<https://doi.org/10.1039/C39900000847>). Moreover, Papers where a methyl groups has been noticed to act as electron rich entity date back to the early 1990ies (mostly studies of interactions with water), none of which was cited in the article:

<https://doi.org/10.1063/1.465050>; <https://doi.org/10.1063/1.466569>;
<https://doi.org/10.1063/1.468280>; <https://doi.org/10.1016/j.cplett.2008.11.009>;
<https://pubs.acs.org/doi/abs/10.1021/jp202704c>;
<https://pubs.acs.org/doi/abs/10.1021/ct300132e>; <https://doi.org/10.1039/C5CP03704B>;
<https://doi.org/10.1007/s12039-015-0861-7>;
<https://doi.org/10.1134/S0036024416100150>;
<https://doi.org/10.1016/j.comptc.2016.06.014>; <https://doi.org/10.1002/cphc.202000927>,
<https://doi.org/10.3390/molecules24183370>,

We would like to thank the referee for pointing out the existence of several papers dealing with sp^3 -C atoms acting as hydrogen bond acceptors, especially focused on the methane-water dimer. We were not aware of these reports and, of course, they must be cited when discussing the existing literature on the topic. The inclusion of such references surely gives a better contextualization of our investigation.

Regarding the Mg...C contact, it is true that the authors mention the existence of this short contact in the original paper from 1990, but it is considered as an agostic interaction rather than a non-covalent Alkaline-earth bond. Here, we present a new rationale to include it within a larger set of interactions in which methyl groups behave as Lewis bases via their carbon atoms. We have added a comment on this respect in the manuscript (page 6).

2. Additional literature references

There are several recent important papers dealing specifically with interactions involving methyl groups that should be cited:

<https://doi.org/10.3390/molecules24183370>;

<https://doi.org/10.3390/c6040060>;

<https://doi.org/10.1002/anie.201811171>

The suggested references are now included in the manuscript.

The reference to tetrel bonding interactions (ref 10) misses the most relevant papers

<https://doi.org/10.1039/C3CP53369G>;

<https://doi.org/10.1002/anie.201306501>;

<https://doi.org/10.1002/tcr.201500256>;

References on tetrel bonding are now included in the article.

3. Tetrahedral distortions

The reported van der Waals corrected distances with the alkali metals are very small and reported to be up to -1.6 \AA . This would imply some covalent character and as a result I would expect that the tetrahedral geometry of a methyl carbon will be distorted. However, this is not reflected in the data: there is no correlation between the van der Waals corrected distance and the shortest E–C–H angle. Do the authors have an explanation for this? I suspect this is also the case for the other datasets.

We are grateful to the referee for raising this point. We missed the impact of the interactions in the geometry of the involved molecules while this is a very important feature of noncovalent interactions with short interatomic distances.

When we looked first at the geometries of the interacting methyl groups we did not see any significant trend. And this is because we were comparing it with an undistorted tetrahedron instead of with the real geometry of the AlMe₃ methyl group when the molecule is alone (no interaction at all), which is in fact slightly distorted towards a vacant octahedron.

The Continuous Shape Measures (CShM) is a very useful tool to calibrate distortions from ideal shapes (polyhedral). We have applied them to the systems under study here and they really helped clarify things. One can build up a Minimal Distortion Pathway to go from a planar trigonal to a vacant octahedron reference shape and measure the CShM value of these to shapes for all model systems in Fig. 8. Then, when represented in a graph, there appear three different groups: a first group with 3a-6a, very close to the undistorted reference shape (blue square); a second group 3b-6b, still close to the reference shape but distorted towards a more planar CH₃ arrangement (TP-3); and finally a third group with 1-2, which is clearly more planar and separated from the original undistorted CH₃ geometry. These groups with different degrees of

geometry changes are in very good agreement with the interaction strength (and, of course, with the corrected interatomic vdW distance).

A figure (Fig. 9) has been added along with a discussion on the geometrical distortions of methyl groups upon interaction. We think that the fact that significant geometrical rearrangements are observed in the interacting methyl groups reinforces the main thesis of the paper.

4. A redo of a CSD search gave different data

In Figure 3 of the paper, the authors present a CSD search for examples of E-CH₃...X structures where the C...X distances are shorter than the van der Waals radii according to Alvarez. They plot the ECX angle as a function of the van der Waals corrected CX distance, revealing that the densest feature is at very short distances and near an ECX angle of 180 degrees. This would be consistent with their idea that methyl-C is a directional 'sigma-hole acceptor'. Intrigued by these data, I wanted to repeat the search but came to very different data when using some looser criteria and when applying the criteria described in the paper. In the data I obtained, the feature near 180 degrees is the minor feature by far. Can the authors explain this discrepancy? The conquest file is included for the authors and the plots are given below.

Figure 3 caption is mistaken, we apologize. Carbon was not considered as E in searches. We have redone the searches without C to check that we get the same results as in original Fig.3 and the data is now presented as a heat map (2D histogram).

In fact, removing C from searches, and only allowing elements that are more electropositive than C, sharpens the tendency for angles closer to 180°, which is in line with the thesis of the article.

Moreover, we have redone Fig. 5a in the absence of C as E and also of B as Y, which was previously seen not to participate in pi-hole interactions in this context (CrystEngComm 2017, 19, 32663).

Fig. 7 has also been modified and it's now presented with Cl as X and without C as E, to be consistent with the rest of the structural analysis.

We think this new presentation of the structural data gives a clearer picture of the bonding topology.

5. Minor issues

- The definition of E is different in Fig.1 and 2 vs 3. This has to be fixed.

There are no energies in Fig.3 and energies in Fig. 1 and 2 are given as Vs.

- Page 2, line 17: '...other electron-rich species had...' should be 'have'.

Fixed.

- The use of personalized language should be modified (e.g. 'we first need to', 'let us look', etc.).

The manuscript has been revised to avoid those expressions.

- Page 3, MEP should be defined as 'molecular electrostatic potential' and EP should be defined.

Corrected.

- Why is the CSD data not plotted as heat plots? I think those are much clearer and more informative because the most dense regions are immediately obvious.

CSD data is now presented as heat maps.

- The term 'nucleophile' is typically used to describe the attacking entity in a chemical reaction and some might thus take issue with its use in the context of an interaction. My advice is to avoid the term.

The term "nucleophile" has been substituted for "Lewis base".

Reviewer #3 (Remarks to the Author):

In the article, the authors try to convince the reader that the methyl group, through its face, can quite often participate in intermolecular interactions with almost any Lewis acid.

This idea is interesting and the method of conducting research is appropriate. However, it seems to me that the authors should be a bit more reserved in this respect and try to defend their thesis more. Although the authors support this idea with 'evidence' from the crystallographic database, it does not convince me entirely.

We would like to thank the referee for his time to read and analyse in detail the manuscript. We have tried to address all his concerns and we strongly believe that the quality of the article has significantly increased. Specially clarifying is the new NCI analysis that complements the QTAIM results.

The doubts are as follows:

1) Figures 3, 5a, and 7b show in fact a considerable angular spread even outside the range of ca. 80-100 degrees.

This is true, but we think it doesn't invalidate our reasoning. Although there is an angular spread, there is a tendency for linear arrangements (in the case of Fig. 3 and 5a and 7a) as the interatomic distance shortens. Out of this behavior, the presence of other dominant interactions is the responsible of the less populated peaks around 80-100° in Figs. 3 and 7a. An explanation has been added to the manuscript trying to shed light on this issue (Page 5).

2) The authors clearly support their idea with the presence of BP, which, however, is not as clear as it might seem. Whether or not BPs should be believed so much has

been extensively debated in the literature and has been perfectly summarized in recent articles by Jablonski, Shahbazian, and Clark, see for example [J. Comput. Chem. 2018 39 2183; ChemistryOpen 2019 8 497; Chem. Eur. J. 2018 24 5401; J. Mol. Model. 2018 24 142] The authors should mention this at least briefly.

We have included a discussion on the reliability of QTAIM for the study of noncovalent interactions, including the references provided by the reviewer, along with a new NCI analysis to enhance the topological study of the electron density.

3) Distances C...Y are rather long.

C...Y distances are shorter than the sum of the van der Waals radii in all studied case, and they can be as short as in models 1 and 2 (1.62 and 1.36 Å shorter than sum of vdW radii, respectively).

4) Instead of talking directly about the C...Y interaction (how to prove its existence?), it is probably better to refer to Me...Y. Then, however, the correct alignment of Me can be considered as a 'geometric combination' of three sigma bond...Y interactions.

We agree with the referee that it is always tricky to talk about the interaction of two particular atoms in the context of complex molecules. Here, we talk indistinctly about C...Y, methyl...Y and methyl...Lewis acid. In any case, we consider that the location of the $V_{s,min}$ (MEP maps), the QTAIM bond paths and the new NCI analysis give enough evidence for speak about a C...Y interaction. All in all, in the title of the article we refer to methyl groups.

Therefore, in this matter, I give priority to the theoretical model systems considered by the authors in the last part of the article. Although this part is probably more convincing,
I feel unsatisfied with the techniques used:

5) QTAIM is troublesome (see comment above)

As mentioned above, topological analysis of the electron density has been expended with a NCI study of both experimental and model systems. The results, presented in the manuscript, are in good agreement with previous QTAIM results and with the general thesis of the article.

6) The result from NBO is rather obvious as the NBO method is structured to give MO_{occ}-> MO_{virt} interactions. The methyl group is closest, so what orbitals other than sigma_{CH} and sigma_{AIC} were supposed to interact?

We know that there is little surprise with the NBO results, but we prefer to keep a paragraph in the manuscript to discuss about them. It is also interesting that for systems 3a-6a the sigma(C-Al) and sigma(C-H) occupied orbital are of similar importance when acting as density donors. Also in the article, interested readers are referred to the Supplementary Information for further details.

7) I think that the dimers investigated are perfect to use the NCI method and I am surprised that this has not been done. The NCI method should elegantly show whether the electron density is mainly located on the C...Y axis or within H...Y. NCI is available in the AIMAll used.

We thank the referee for pointing about the usefulness of this method. A comprehensive NCI analysis has been performed for all studied systems and the results are presented and discussed in the main text.

8) The authors could think of still other methods showing electron density localization (ELF, Laplacian or L, etc.), although some may require another programs.

We have analysed the Laplacian of the electron density but the results are not conclusive so we have decided not to include them in the paper. If the reviewer and/or the editor think is worth showing them, they could be included in the Supplementary Info.

On the other hand, being honest we lack the expertise to undertake ELF analysis and we are not sure it would give new insight into the investigation.

Nevertheless, in summary, I believe that the present article deals with an interesting issue and may cause a broader debate among chemists, so it should be published. However, before that, the authors should consider the above remarks (especially describe the allegations regarding QTAIM) and supplement the theoretical analysis with, at least, the NCI calculations.

As the correction requires additional calculations, my grade is MAJOR corrections.

Other (minor) comments go only to the authors.

Minor:

2/7 & 12/6: should be "occupied orbital"

Fixed.

2/13: should be "Lewis acids"

Fixed.

2/22: "for instance" should be removed because, according to the accepted nomenclature, 'acidic' C always forms tetrel bonds

Fixed.

2/23: Carbon atoms cannot be "in the form of carbenes".

Fixed.

2/26: "... density donors." - references should be added

Reference has been added.

2/31: "we can ask ourselves"; stylistically awkward, maybe, e.g. 'we may ask if ...'

"we can ask ourselves" has been substituted for "a question arises".

3/7: molecular electrostatic potential (MEP)

Fixed.

3/12: "leading to"; better would be ", that is ..."

Fixed.

3/6&13: The use of the expressions "tetrasubstituted" and "substituted" is confusing because it is difficult to say that in the group -CH₃ a carbon atom is substituted, especially with four substituents.

The term tetrasubstituted has been eliminated.

7: The plot for the C...Y-R angle should also be shown because 5a alone is not too convincing.

The suggested plot, along with a comment on it, has been added to Figure 5.

11: Dimer 2 is missing in Figure 8.

Dimer 2 is present in Fig. 8, I do not know if it's missing for the referee for some reason...

11/3&7: should be "selenium atoms" and "C...Se bond paths", respectively

Fixed.

Table 1: The authors should rather report absolute distance than vdW-corrected ones, especially in the theoretical part (Table 1).

We think that vdW-corrected distances are useful because they allow comparison no matter what the atoms involved are. In any case, absolute interatomic distances have been added to Table 1.

12/17: "demonstrated...physical origin ... is electrostatic" - The authors should clearly state how they think they have found this electrostatic origin.

The MEP maps evidence that the interactions must have an electrostatic component. Moreover, the results from our model systems (Fig. 8) show that making the sigma-hole more positive at Y (by replacing CH₃ with CF₃) reinforces the interaction, which is also indicative of an electrostatic origin. An explanation at this respect has been added to the manuscript (Last conclusions paragraph)

13/2: "The high directionality" - this seems to be somewhat exaggerated

It has been rewritten as "significant directionality".

REVIEWER COMMENTS

Reviewer #1 (Remarks to the Author):

The authors have adequately addressed the comments and suggestions made by me, and in my opinion also those of the other two reviewers. Publication of the current version is recommended.

Reviewer #2 (Remarks to the Author):

The authors addressed my earlier points in a satisfactory manner and the paper is thus suitable for publication.

Reviewer #3 (Remarks to the Author):

The authors have significantly improved the article that can be published after taking into account 3 minor comments:

Minor:

line 36: should be "acting as Lewis acids"

line 45: The Authors may also cite [Alkorta, Elguero, J. Phys. Chem. 1996, 100, 19367; Jablonski, Palusiak, Phys.Chem.Chem.Phys. 2009, 11, 5711; Jablonski, Molecules 2021, 26, 2275]

line 443: Correct Chal/asiński to Chałasiński or just Chalasiński

Mirosław Jablonski

REVIEWERS' COMMENTS

Reviewer #1 (Remarks to the Author):

The authors have adequately addressed the comments and suggestions made by me, and in my opinion also those of the other two reviewers. Publication of the current version is recommended.

Reviewer #2 (Remarks to the Author):

The authors addressed my earlier points in a satisfactory manner and the paper is thus suitable for publication.

Reviewer #3 (Remarks to the Author):

The authors have significantly improved the article that can be published after taking into account 3 minor comments:

Minor:

line 36: should be "acting as Lewis acids"

line 45: The Authors may also cite [Alkorta, Elguero, J. Phys. Chem. 1996, 100, 19367; Jablonski, Palusiak, Phys.Chem.Chem.Phys. 2009, 11, 5711; Jablonski, Molecules 2021, 26, 2275]

line 443: Correct Chal/asiński to Chałasiński or just Chłasiński.

The three points raised by reviewer #3 have been addressed.